# Caffeine Increased Muscle Endurance Performance Despite Reduced Cortical Activation and Unchanged Neuromuscular Efficiency and Corticomuscular Coherence

**DOI:** 10.3390/nu11102471

**Published:** 2019-10-15

**Authors:** Paulo Estevão Franco-Alvarenga, Cayque Brietzke, Raul Canestri, Márcio Fagundes Goethel, Bruno Ferreira Viana, Flávio Oliveira Pires

**Affiliations:** 1Exercise Psychophysiology Research Group, School of Arts, Sciences and Humanities, University of São Paulo, São Paulo 03828-000, Brazil; 2Physical Education, Estácio de Sá University, Resende, Rio de Janeiro 27515-010, Brazil; 3Rehabilitation Sciences Graduate Program, Augusto Motta University Center, Rio de Janeiro 21041-010, Brazil

**Keywords:** fatigue, placebo, ergogenic, EEG–EMG coherence

## Abstract

The central and peripheral effects of caffeine remain debatable. We verified whether increases in endurance performance after caffeine ingestion occurred together with changes in primary motor cortex (MC) and prefrontal cortex (PFC) activation, neuromuscular efficiency (NME), and electroencephalography–electromyography coherence (EEG–EMG coherence). Twelve participants performed a time-to-task failure isometric contraction at 70% of the maximal voluntary contraction after ingesting 5 mg/kg of caffeine (CAF) or placebo (PLA), in a crossover and counterbalanced design. MC (Cz) and PFC (Fp1) EEG alpha wave and vastus lateralis (VL) muscle EMG were recorded throughout the exercise. EEG–EMG coherence was calculated through the magnitude squared coherence analysis in MC EEG gamma-wave (CI > 0.0058). Moreover, NME was obtained as the force–VL EMG ratio. When compared to PLA, CAF improved the time to task failure (*p* = 0.003, d = 0.75), but reduced activation in MC and PFC throughout the exercise (*p* = 0.027, d = 1.01 and *p* = 0.045, d = 0.95, respectively). Neither NME (*p* = 0.802, d = 0.34) nor EEG–EMG coherence (*p* = 0.628, d = 0.21) was different between CAF and PLA. The results suggest that CAF improved muscular performance through a modified central nervous system (CNS) response rather than through alterations in peripheral muscle or central–peripheral coupling.

## 1. Introduction

Caffeine is one of the most widely ergogenic aids traditionally used to improve physical performance in different exercise scenarios [1,2] such as team sports [3], cycling exercise [4,5], and muscular function tests [6,7,8]. However, the underlying mechanism of caffeine ingestion on either whole-body or muscular endurance exercise performance is still controversial, as it involves central and peripheral hypotheses such as alterations in central nervous system (CNS) and skeletal muscles, respectively. It has been well known that caffeine inhibits A1 adenosine receptor and postsynaptic A2a receptor in CNS [9,10] and muscles [7,8], thereby, improving spinal and supraspinal excitability and altering cortical and muscular activation during exercise. Nevertheless, it is still debatable whether caffeine improves endurance performance through coupled or uncoupled alterations in both central and peripheral sites. 

Results of studies using micromolar doses of caffeine suggest that alterations in CNS are the likely mechanism of the caffeine effects on endurance performance. Studies have shown an increased spinal and supraspinal excitability with caffeine [11,12], thus justifying the increased muscular endurance performance as measured as the time to task failure at a submaximal target force [13]. Interestingly, an earlier study observed that caffeine ingestion (6 mg·kg^−1^ of body mass) reduced the motor-related cortical potential at the vertex during a submaximal isometric knee extension. The authors concluded that caffeine decreased the magnitude of excitatory inputs from frontal and primary motor cortex (MC) areas necessary to produce a given force, likely due to an enhanced spinal and supraspinal excitability [14]. Based on these arguments, one may expect that less activation in MC and frontal cortex areas would be required to sustain a target force after caffeine ingestion, thus improving the muscular endurance capacity as measured by time to task failure. Unfortunately, that study used a closed-loop isometric exercise (i.e., 4 × 10 muscle contractions), so that time-to-task failure measures were not provided. Moreover, motor-related cortical potential measure may be indicative of readiness (i.e., excitability) rather than activation, thus cortical electroencephalography (EEG) measures throughout a time-to-task failure exercise may be insightful for this proposal. In fact, an earlier study found a reduced cortical EEG alpha wave after caffeine ingestion at rest [15] so that a study exploring the cortical EEG alpha wave during exercise is yet to be provided.

On the other hand, studies have suggested that caffeine may also improve endurance exercise performance through an enhanced muscular function. For example, an earlier study had observed that caffeine increased tetanic force stimulated at 20 Hz but not at 40 Hz [7]. Additionally, another study verified that caffeine (6 mg·kg^−1^ of body mass) increased biceps brachii electromyography (EMG) and maximal isokinetic force of elbow flexion at different angular velocities [8]. Somehow, caffeine may have also improved neuromuscular efficiency as caffeine ingestion increased muscle fiber conduction velocity. Consequently, beyond the reduced cortical activation one may argue that caffeine ingestion improves muscular endurance capacity through an ameliorated neuromuscular efficiency. However, evidence that caffeine may improve muscular endurance (i.e., time to task failure) together with a reduced cortical activation and increased neuromuscular efficiency has yet to be provided in a single study design.

There is a paucity of studies simultaneously investigating the caffeine effects on both central and peripheral responses to a muscular endurance performance. For example, an earlier study verified that the increased maximal voluntary force during maximal knee extensions after caffeine ingestion was associated with alterations in central more than in peripheral responses [16]. In contrast, a recent study observed that improved single-leg knee extension performance after caffeine ingestion was associated with ameliorated central and peripheral fatigue indexes [12]. Therefore, studies simultaneously investigating central and peripheral responses to caffeine ingestion during muscular endurance performance are insightful to reveal the importance of central and peripheral effects of caffeine.

In a central vs. peripheral fatigue scenario, it is still unknown whether caffeine improves muscular endurance performance through a coupled alteration in central and peripheral locations, as one may argue that caffeine could improve exercise performance through independent effects on cortical and muscle responses. Analysis of the strength of corticomuscular coupling during a time-to-task failure protocol may be helpful to understand how caffeine affects the link of information being processed in these two different locations [17]. In this regard, analysis of the EEG–EMG linear dependency in time and frequency domains indicates the neuronal synchronicity between cortical and muscle activation [18], so that EEG–EMG coherence during a time-to-task failure protocol may provide insights into caffeine effects on corticomuscular coupling and fatigue. Unfortunately, possible caffeine effects on fatigue and EEG–EMG coherence relationship remain uninvestigated.

The present study verified whether increases in endurance performance after caffeine ingestion occurred together with changes in cortical activation, neuromuscular efficiency, and EEG–EMG coherence. Based on independent results, we hypothesized that caffeine ingestion would reduce cortical activation and increase neuromuscular efficiency, thereby increasing the time to task failure during single-leg knee extension protocol. Additionally, a likely improved coherence between central (i.e., MC EEG) and peripheral sites was expected with caffeine ingestion.

## 2. Materials and Methods 

### 2.1. Participants

A sample size of 10 participants was determined, having a significance level of 5%, a power >0.95, and an effect size (ES) >0.8 (G-Power software, version 3.1., Dusseldorf, Germany). However, we expected a 20% dropout so that 12 participants volunteered to participate in this study. Thus, recreationally trained cyclists (34.3 ± 6.2 years old; 179.3 ± 5.1 cm; 77.6 ± 6.8 kg), non-smokers and free from cardiovascular, visual, auditory, and cognitive disorders were recruited. Briefly, three were non-consumers (≤40 mg of caffeine per day), five were occasional consumers (≤250 mg of caffeine per day), and four were daily consumers of caffeine (250 < consumption < 572 mg of caffeine per day), according to classification proposed elsewhere [1,19]. Importantly, caffeine has been suggested as an ergogenic aid capable of improving endurance performance, regardless of habitual caffeine consumption [5,20]. They were oriented to avoid consumption of coffee or any stimulant (energy drink, etc.) and alcoholic beverages, as well as intense exercise for 48 h preceding the sessions. Experimental procedures, risks, and benefits were explained before collecting their written consent form signature. The procedures were previously approved by a local Ethics Committee (Process: 63787816.1.0000.5390) from the University of São Paulo and performed according to the Declaration of Helsinki.

### 2.2. Study Design

The design of the present study involved five sessions. During the sessions 1 and 2, participants were familiarized with instruments and procedures of knee isometric extension (IC) and EMG and EEG measures. Moreover, participants performed three maximal voluntary contractions (MVC) and a submaximal IC to task failure set at 70% MVC. These procedures were repeated during session 2, and the force attained in MVC was adopted to determine the IC intensity (i.e., 70% MVC) used in the following sessions. Session 3, baseline trial (CON): Participants performed a baseline IC to task failure with no supplementation; sessions 4 and 5, supplementation trials: Participants performed a submaximal IC exercise ~45 min after caffeine (CAF) or placebo (PLA) ingestion. Sessions 1, 2, and 3 were performed in sequential order, as we were interested in properly familiarizing participants with procedures before assessing EEG, muscular efficiency, and EEG–EMG coherency in baseline submaximal IC. Then, we performed sessions 4 and 5 in a double-blinded, counterbalanced order as we intended to investigate central and peripheral responses to IC exercise after caffeine ingestion. Therefore, the baseline session was used as a familiarization when assessing physiological responses to a “natural” non-supplemented IC exercise. The study was finished within 30 days. The sessions were interspersed by a 3–7 day washout period, being performed at the same time of the day in a controlled environment (∼24 °C and 50%–60% humidity). This experimental setup was part of an umbrella research project that studied caffeine effects on several psychophysiological responses to different exercise modes. Importantly, experimental procedures used in other parts of the umbrella project that have been already published [4] are unlikely to influence the outcomes measured in the present study [20]. Hence, with the exception of the ingested substance, all experimental trials (CON, CAF, and PLA) were conducted under identical and controlled conditions, thus ensuring the reliability of the present study.

### 2.3. Caffeine and Placebo Ingestion

Participants ingested 5 mg·kg^−1^ of body mass of caffeine 45 min before the submaximal IC to task failure. This is in accordance with recommendations of the International Society of Sports Nutrition (ISSN) for caffeine ingestion [1], suggesting that 3–6 mg·kg^−1^ of body mass of caffeine significantly improves endurance performance when ingested from 45 to 60 min before the exercise bout [1]. Caffeine and placebo were manipulated in capsules of the same size, color, and smell so that participants and the researcher directly involved in data sampling were unaware about the substance ingested. Participants received a capsule containing CAF or PLA (lubricant, magnesium stearate, and magnesium silicate) in a typical double-blind trial, having 50% chance of ingesting the actual active or placebo substance. The blinding efficacy was checked after the participants finished their participation.

### 2.4. MVC and Isometric Contraction to Task Failure

Initially, participants were accommodated in a custom-built single-leg knee extension chair attached to a cell load (EMG System^®^, São José dos Campos, Brazil) to measure a force of 2 kHz frequency, having their hips and knees at 90° and 60° from the horizontal axis, respectively. Their chest and hips were carefully fixed in order to avoid accessory movements. After familiarizing with the MVC and IC protocols in session 1, participants repeated them in session 2. Moreover, in session 2 participants performed three sets of three MVC (interspersed by a 3 min interval) in order to assess the highest peak force value between them and subsequently determine the submaximal IC workload. The IC protocol consisted of performing an isometric knee extension to task failure at 70% MVC. Therefore, after a warm-up consisting of unloaded squats (two sets of 15 repetitions with 1 min interval between sets), participants sat on the chair which was individually adjusted. They were oriented to maintain the force corresponding to 70% MVC (±5% variation) by using a visual feedback on a computer screen. The task failure was identified as the inability to maintain the target force after three verbal encouragements [21]. Measures of force (expressed as kgf), EEG, and EMG were recorded throughout the submaximal IC. 

### 2.5. Measures and Instruments

#### 2.5.1. Electroencephalography (EEG)

Activation in MC and PFC was continuously obtained through an EEG unit (Emsa^®^, EEG BNT 36, TiEEG, Rio de Janeiro, Brazil) at Cz and Fp1 position, respectively, according to the international EEG 10–20 system [22]. These positions were ensured according to frontal and sagittal planes, referenced to the mastoid. The EEG was recorded at a 600 Hz sampling frequency, through active electrodes (Ag–AgCl) with resistance ~5 KΩ. After exfoliation and cleaning, electrodes were fixed with a conductive gel, adhesive tape, and medical strips. The EEG signal was recorded during a 3 min baseline before CAF or PLA ingestion (when participants were completely calm) as well as throughout the submaximal IC. They were oriented to avoid head and trunk movements during baseline and exercise phases. 

An EEG signal with amplitude >100 µV was considered as an artifact (*n* = 1–2, depending on the moment of the experimental setup) and removed from analysis [23]. In baseline EEG data, data recorded during the first and last 30 s of a 180 s time window were removed (to avoid noise associated with the increased expectation of the start and stop of EEG sampling) and a fast-Fourier transformation calculated the total power spectral density (tPSD) within 8–13 Hz (alpha wave) over the most steady (i.e., lowest standard deviation (SD)) 30 s time window. In exercise EEG data, a fast-Fourier transformation calculated the tPSD within the alpha wave over the last 2 s of every 25% of the submaximal IC duration (i.e., 25%, 50%, 75%, and 100%), thereafter the exercise EEG data were expressed as a percentage of the baseline. Importantly, we used the EEG alpha wave to indicate activation as this EEG frequency is suggested to reflect an increased number of neurons coherently activated [24] as indicated by the increase in inhibited neurons–to–disinhibited neurons relationship [25]. In this regard, an increased alpha wave may indicate a cooperative-synchronized behavior of a large number of activated neurons [25]. All EEG analyses were performed through an algorithm in Matlab^®^ environment.

#### 2.5.2. Neuromuscular Efficiency (NME)

Initially, participants had their skin shaved, exfoliated, and cleaned with isopropyl alcohol to reduce the skin impedance. Thereafter, a bipolar electrode was placed over the belly of the vastus lateralis muscle (VL) according to the probable muscle fiber orientation. The EMG signal was recorded throughout the submaximal IC through an EMG unit (EMG System, São José dos Campos, Brazil) at a 2 kHz sample rate (gain 1000) with a recursive fourth-order Butterworth bandpass filter (cutoff frequencies between 20 and 500 Hz), before calculating the root-mean-square value (RMS) of the EMG signal. All EEG data collection followed the Surface Electromyography for the Non-Invasive Assessment of Muscles standards [26].

The neuromuscular efficiency (NME) was obtained as the force–EMG RMS ratio of the EMG burst over the last 2 s of every 25% of the submaximal IC duration, as proposed elsewhere [27]. Importantly, as a reduction in NME is expected as fatigue progresses, indicating that more motor units have been recruited to produce the same force, NME has been suggested as a peripheral fatigue index [27]. Hence, to obtain the NME, we also filtered force data through a recursive fourth-order Butterworth low-pass filter, having a cutoff frequency determined by residual analysis at 7 Hz, before normalizing force data by body mass. Thereafter, the NME index (expressed as arbitrary units) was calculated as the integral of the force–EMG RMS relationship over a 250 ms time window with a 249.5 ms overlap (Equation (1)).

(1)$$NME=\sum \overset{i+1}{\underset{i}{\int }}\frac{Force_{\left(i\right)}}{\sqrt{1/500\cdot {\left(x_{i}^{2}+x_{i+1}^{2}+\ldots +x_{i+499}^{2}\right)}_{\left(i\right)}}}$$

#### 2.5.3. EEG–EMG Coherence

Initially, we checked through a 95% confidence interval (CI) calculation which EEG spectral wave from MC (Cz position) revealed coherence with VL EMG signal, as suggested elsewhere [28]:(2)$$CL=1-0.05^{1/n-1}$$
where *n* is the number of windows used for spectral estimation. Given the varied time to task failure, the number of windows was not the same for all spectral estimates.

Afterward, we computed the power spectral of the rectified EMG and EEG gamma wave (30–50 Hz) through Welch’s method, having a 50% overlapped Hamming window with 512 samples in each section. Only active data (i.e., between onset and offset of each trial) were used to calculate the power spectral, and the magnitude squared coherence between EEG and EMG (expressed as arbitrary units) was then obtained:(3)$$coh_{c1,c2}\left(f\right)\;=\;\frac{{\left|S_{c1c2}\left(f\right)\right|}^{2}}{S_{c1c1}\left(f\right)\cdot S_{c2c2}\left(f\right)}$$
where *Sc*_1_*c*_1_ and *Sc*_2_*c*_2_ are the auto-spectra of each signal; *Sc*_1_*c*_2_ is the cross-spectra. Accordingly, EEG–EMG coherence data were calculated at each 25% of the submaximal IC duration. 

#### 2.5.4. Statistical analyses

Results were reported as mean and standard deviation (±SD). Firstly, one-way ANOVA (Bonferroni as a post hoc) was used to compare muscle endurance performance (expressed as time to task failure in submaximal IC) in baseline, CAF, and PLA conditions. Additionally, we also expressed endurance performance as a percentage of alteration from the baseline session, thus comparing CAF and PLA trough a paired T-test. Secondly, MC and PFC activation (indicated by EEG alpha wave), NME, and EEG–EMG coherence responses to submaximal IC between CAF and PLA conditions were compared at every 25% of the total exercise duration through a 4 × 2 mixed model, having time (25%, 50%, 75%, and 100%) and ingestion (CAF vs. PLA) as fixed factors and participants as the random one. The AIC index (Akaike’s information criterion) determined the covariance matrix that best fitted to the dataset (homogeneous and heterogeneous compound symmetric, first-order auto-regressive, auto-regressive moving average, and Toeplitz), while Bonferroni test was used in multiple comparisons.

We reported the post hoc ES analysis (expressed as d-Cohen) as a qualitative analysis approach, so that ES was interpreted as small (<0.2), moderate (0.2–0.6), large (0.6–1.2), very large (1.2–2.0), and extremely large (>2.0), as suggested elsewhere [29]. Results were significant when *p* < 0.05.

## 3. Results

### 3.1. Baseline Session and Blinding Efficacy

Participants attained a task failure in 28.5 ± 16.4 s in baseline session. In order to check the blinding of manipulation, participants were asked to guess which substance they thought they ingested in each session. In total, nine participants correctly identified CAF (and consequently PLA) ingestion, while three did not. Participants reported no adverse effects from caffeine ingestion.

### 3.2. Caffeine Effects on Muscle Performance

A condition main effect was found (F = 8.489; *p* = 0.002; d = 1.242 very large ES) so that the absolute time to task failure was greater in CAF than in PLA (0.007) and baseline (0.006). When expressed as a percentage of alteration from baseline session, CAF (33.5 ± 14.2 s; (95% CI = 23.9, 40.2), 9.1% ± 36.4% from baseline) further improved muscular endurance performance (t = 3.993, *p* = 0.003, d = 0.75 large ES) when compared to PLA ingestion (25.8 ± 10.6 s; (95% CI = 18.6, 30.8), −7.7% ± 25.1% from baseline) (Figure 1).

### 3.3. Caffeine Effects on Central and Peripheral Indexes

Regarding MC activation, EEG Cz activity was significantly reduced (F = 5.654, *p* = 0.027, d = 1.01 large ES) when compared to PLA. Furthermore, a moment main effect was observed as MC activation increased throughout the exercise (F = 3.767, *p* = 0.025, d = 0.83 very large ES). In contrast, no ingestion by moment interaction effects were found (F = 2.462, *p* = 0.125, d = 0.67 large ES). Accordingly, an ingestion main effect (F = 4.925, *p* = 0.045, d = 0.946 large ES) as well as a moment main effect (F = 10.360, *p* = 0.001, d = 1.37, very large ES) was observed in PFC activation, as CAF reduced the EEG Fp1 activity when compared to PLA, although PFC activation has increased throughout the submaximal IC exercise. Moreover, no ingestion by moment interaction effects was found (F = 1.280, *p* = 0.343, d = 0.48, moderate ES). Figure 2 shows these EEG results.

Regarding the NME results, caffeine ingestion was ineffective in improving VL muscle efficiency when compared to PLA (F = 0.065, *p* = 0.802, d = 0.34 moderate ES). However, a moment main effect was detected as NME changed throughout the exercise (F = 7.97, *p* < 0.001, d = 1.20 very large ES). Additionally, no ingestion by moment interaction effect was observed (F = 0.006, *p* = 1.00, d = 0.02 small ES). Figure 3 depicts these results.

Previous analysis revealed that EEG–EMG coherence was significant (CI > 0.0058) in EEG gamma wave, as shown by spectrograms (Figure 4). We observed that neither CAF session (F = 0.240, *p* = 0.628, d = 0.21 moderate ES) nor moment main effect (F = 0.437, *p* = 0.727, d = 0.28 moderate ES) changed EEG–EMG coherence. Accordingly, we did not observe ingestion by moment interaction effects (F = 0.522, *p* = 0.670, d = 0.35 moderate ES) in EEG–EMG coherence, as shown in Figure 5. Table 1 presents mean (±SD) and 95% confidence interval (95% CI) values of dependent variables. 

## 4. Discussion

The present study aimed to verify whether an increased time to task failure with CAF ingestion would be followed by changes in cortical activation, neuromuscular efficiency, and EEG–EMG coherence during a single-leg knee extension exercise. Our findings showed that caffeine improved endurance performance, despite a reduced activation in both PFC and MC and unaltered neuromuscular efficiency and EEG–EMG coherence. These results suggested that caffeine ingestion improved muscular endurance performance through located modifications in the CNS rather than alterations in peripheral muscle. Importantly, this is the first study showing that caffeine effects on CNS were uncoupled from peripheral responses.

Different studies have indicated that caffeine effects on A1 adenosine receptor and postsynaptic A2a receptor in CNS are the most likely mechanism underlying improvements in endurance performance [9,10,30]. We hypothesized that caffeine may improve muscular endurance performance in a time to task failure regardless of a reduced activation in PFC and MC as reported elsewhere [15]. Although the increase in PFC and MC activity was a main exercise effect, there was a reduced PFC and MC activation throughout the submaximal IC protocol in the CAF session. A likely explanation of this reduced cortical activation during exercise is an increased CNS excitability with caffeine ingestion, as suggested [10] and confirmed elsewhere [11]. Caffeine has been suggested to increase both the corticospinal [31] and spinal excitability [11], thereby, leading to less excitatory input from frontal to vertex areas as well as from vertex to peripheral muscles, when generating the same force or power output [14,30]. Although we have not measured CNS excitability responses in the present study, the fact that participants maintained the same force requiring less PFC and MC activation throughout most of the submaximal IC exercise may be suggestive of an increased CNS excitability after caffeine ingestion. Thus, as a result of the lower cortical activation necessary to produce a given force, participants may have been capable of further increasing the time to task failure in the CAF session. Interestingly, one may argue that a likely increased spinal and corticospinal excitability promoted by caffeine ingestion extended the time to reach a “cortical activation limit”, as both PFC and MC activation were lower in CAF than PLA from the beginning to 50% and 75% of the exercise duration, respectively, matching a “maximal cortical activation” (as recorded in the PLA session) only at 100% of the IC exercise. However, this suggestion must be interpreted with caution, as this argument is based on visual more than statistical analysis.

Despite the decreased NME as a main exercise effect, caffeine was ineffective in improving muscle efficiency during submaximal IC exercise, so that the improved muscular endurance performance in the CAF session cannot be related to peripheral responses. Controversial results of caffeine ingestion on peripheral responses have been reported. For example, some have found positive caffeine effects either on calcium release from the sarcoplasmic reticulum [32,33] or muscle fiber conduction velocity [8], thereby supporting the notion of a caffeine ergogenic effect on peripheral muscles. However, others have failed to find positive caffeine effects on peripheral muscle indexes such as a peripheral silent period [31] or M-wave [7], suggesting that sarcolemma excitability and tubule T propagation are unaffected by caffeine. In the present study, we observed that caffeine ingestion was ineffective in enhancing neuromuscular efficiency calculated as the force–EMG RMS ratio, thus, indirectly suggesting no caffeine effects on muscle properties. Such an ineffectiveness of caffeine in improving peripheral responses may be related to the muscle contraction stimulation frequency of the submaximal IC exercise, as it has been proposed that caffeine changes muscle properties through alterations in calcium release rather than through potassium accumulation [33]. Thus, assuming that calcium metabolism is associated with force losses mainly in frequencies <20–30 Hz [7,34,35] and that our submaximal IC exercise required a muscle contraction frequency mostly higher than 50 Hz [36], perhaps caffeine ingestion is ineffective in improving key muscular properties enhance muscle endurance performance during submaximal IC. Somehow, the fact that coherence analysis indicated a significant coupling between gamma wave EEG (30–50 Hz) and EMG may reinforce this argument.

Despite studies proposing EEG–EMG coherence analysis as a tool to investigate the corticomuscular coupling between motor cortex and pooled motor units [37,38,39], only a few have been designed to investigate the EEG–EMG coherence and fatigue relationship [18,39]. Coherence, defined as a spectral power covariance between signals from different origins, may provide an estimation of the corticomuscular coupling [39] signals. In a muscle fatigue scenario, EEG–EMG coherence is expected to decrease as exercise progresses, thus suggesting a corticomuscular desynchronization with fatigue [18]. Unexpectedly, we found no main exercise effects on EEG–EMG coherence. Perhaps, the fact that we used a constant rather than an intermittent muscle contraction during submaximal IC can be related to this unaltered coherence during exercise [38], given the less-complex muscle recruitment strategy in this mode of contraction [40,41]. Importantly, the present study was the first to provide evidence that caffeine ingestion maintained the corticomuscular coupling between the motor cortex and pooled motor units in submaximal IC exercise, despite reductions in MC activation and increased muscle endurance. Somehow, the likely increase in corticospinal and spinal excitability in the CAF session [30] may have been associated with a longer sustained force output despite the reduced MC activation, as the signal from MC areas remained coupled with the signal at pooled motor units, even though the progressive fatigue in the CAF session.

## 5. Methodological Aspects, Strength, and Limitations

Beta and gamma waves have been suggested for EEG–EMG coherence analysis of isometric and isotonic muscle contractions, respectively [38,40]. An earlier coherence study observed a greater EEG gamma wave coherence with EMG signal in isometric knee extension, although both EEG beta and gamma frequencies were significantly coherent with EMG [38]. In the present study, we first verified which EEG waves from MC would best reveal coherence with peripheral muscle during IC exercise. In contrast to earlier results [38,40], we found significant coherence in EEG gamma wave during isometric exercise. Perhaps, the fact that our participants had to sustain a target force throughout the exercise until the task failure, while fatigue progressed, may have induced an increase in the median frequency of the motor command to peripheral muscles, thus shifting the coherence toward higher EEG frequencies such as the gamma wave. Additionally, the fact that our participants had to focus on visual feedback on the screen (i.e., horizontal lines delimiting the target force) during the submaximal IC exercise may have also led to a shift toward higher EEG frequencies, as EEG–EMG coherence can occur at higher EEG frequencies when individuals modulate the target force through visual feedback [41].

Importantly, results of the check of blinding efficacy challenged the use of traditional placebo-controlled clinical trials, as suggested elsewhere [42]. Agreeing with previous results [43], we observed that nine out of 12 participants correctly guessed when caffeine was ingested, despite using a typical double-blind, placebo-controlled design. Therefore, one may argue that some of the endurance performance improvements in CAF session may have been potentiated by the expectation of ingesting caffeine, as a recent study verified that placebo perceived as caffeine improved cycling performance as much as caffeine [44]. As recently recommended, future research must take caffeine expectancies into account when investigating caffeine effects on performance [42].

Although we have included participants with different caffeine habituation in the present study, responsiveness and habituation were seemingly not an issue in the present results, as a recent study by Del Coso et al. [20] observed that different individuals responded to caffeine ingestion improving aerobic and anaerobic cycling performance from 9% to 1% across multiple testing sessions and Wilk et al. [45] found ergogenic effects of caffeine ingestion in athletes habitually using caffeine. Moreover, a well-controlled study by Goncalves et al. [5] verified that habitual caffeine consumption did not influence its potential ergogenic effect. Therefore, together these studies reinforce the notion that caffeine consumption habituation had no impact on results of the present study.

Only a few studies have simultaneously investigated central and peripheral effects of caffeine on exercise performance, mainly in a well-controlled design [12,16]. In this regard, the present study contributes to the improvement of the available literature as we showed that caffeine potentiated muscular endurance performance through central rather than through peripheral effects. Importantly, this is the first study showing that caffeine effects on CNS were uncoupled with muscle responses, as we found no effects of caffeine ingestion on corticomuscular coupling. This may be of value for exercise performance and clinical scenarios, as one may want to focus on CNS alterations without altering CNS–muscle coupling or muscle responses. However, an obvious limitation is that caffeine may play a role in multiple physiological responses beyond the electrophysiological ones investigated in the present study, thus caution is needed when inferring caffeine effects on other physiological responses such as tissue oxygenation and cell metabolism.

## 6. Conclusions

Results of the present study showed that caffeine improved muscle endurance performance, regardless of reductions in both PFC and MC activation and unaltered neuromuscular efficiency and EEG–EMG coherence. These results may suggest that caffeine ingestion improved performance in isometric contraction through a modified CNS response rather than through alterations in peripheral muscle or central–peripheral coupling.

## Figures and Tables

**Figure 1 nutrients-11-02471-f001:**
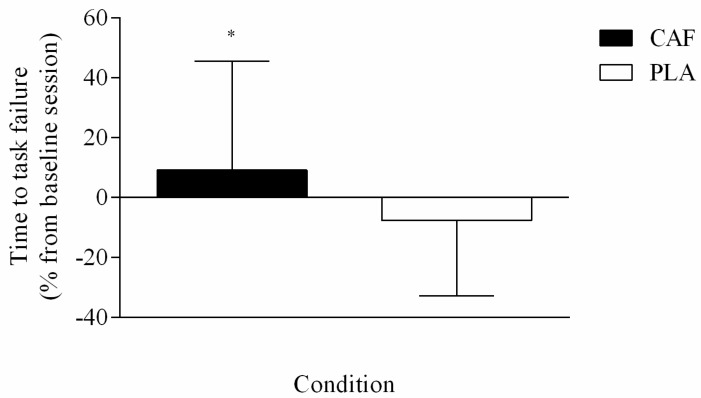
Changes in time to task failure during submaximal isometric contraction after caffeine (CAF) and placebo (PLA) ingestion. * Indicates significantly different from PLA (*p* = 0.003). Data are reported as mean ± standard deviation (SD).

**Figure 2 nutrients-11-02471-f002:**
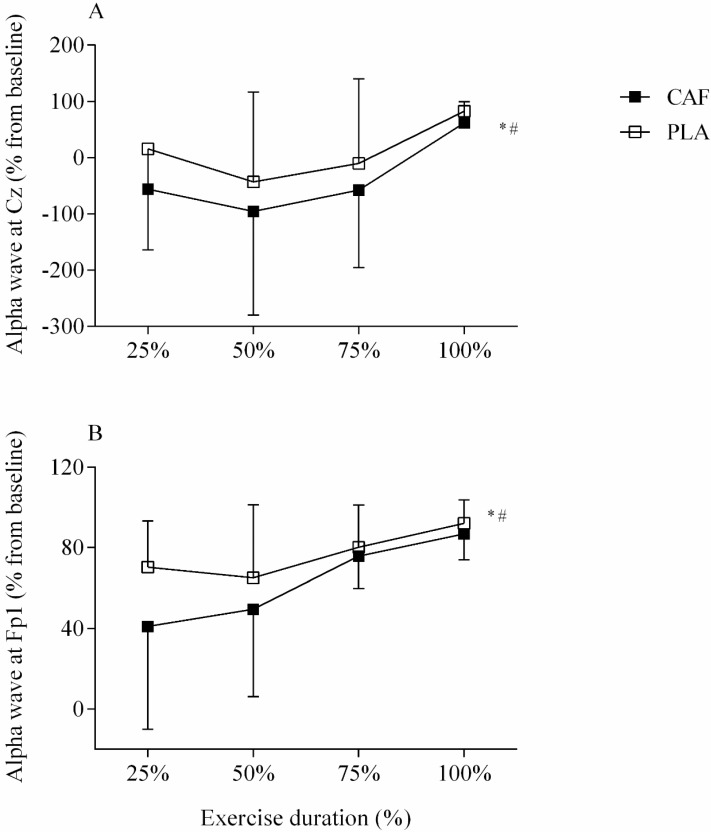
Electroencephalography (EEG) alpha wave recorded at Cz (**A**) and Fp1 (**B**) positions during isometric contraction in CAF (filled square) and PLA (open square) sessions. * Indicates condition main effect in Cz (*p* = 0.027) and PFC (*p* = 0.045). # indicates moment main effect in Cz (*p* = 0.000) and Fp1 (*p* = 0.001). Data are reported as mean ± SD.

**Figure 3 nutrients-11-02471-f003:**
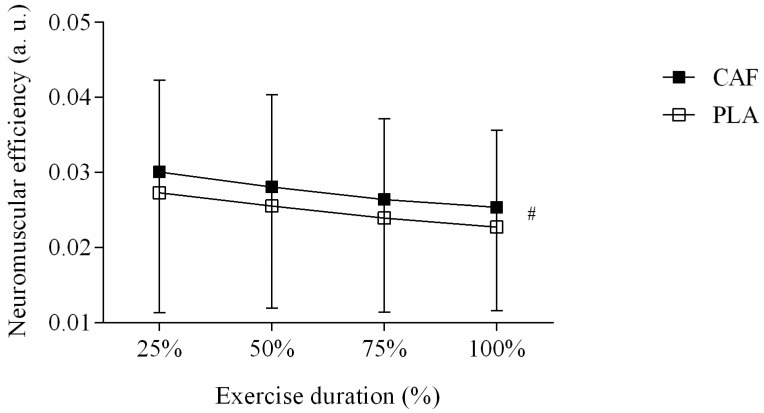
Changes in neuromuscular efficiency (NME) of vastus lateralis muscle during isometric contraction in caffeine (CAF) and placebo (PLA) sessions. # Is moment main effect (*p* = 0.000). Data are reported as mean ± SD.

**Figure 4 nutrients-11-02471-f004:**
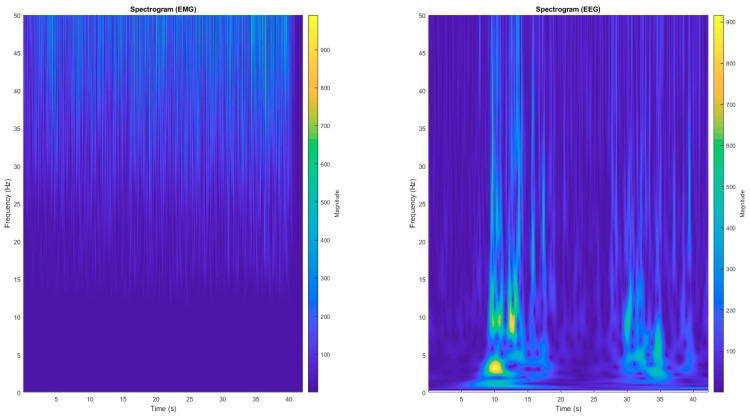
Spectrogram of EEG gamma wave at Cz position and vastus lateralis electromyography (EMG).

**Figure 5 nutrients-11-02471-f005:**
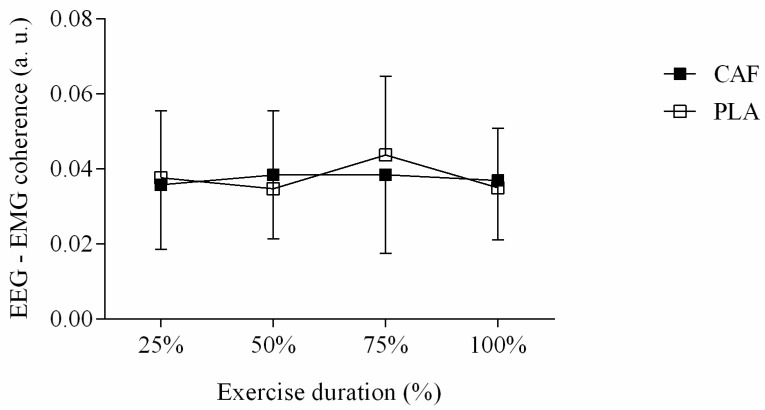
EEG–EMG coherence in CAF (filled square) and PLA (open square) sessions. Data are reported as mean ± SD.

**Table 1 nutrients-11-02471-t001:** Mean (±SD) and 95% confidence interval (95% CI) for dependent variables.

		Time of Exercise			
Ingestion	Dependent Variable	25%	50%	75%	100%
**CAF**	Prefrontal EEG	40.9 ± 54.0	49.4 ± 43.2	75.7 ± 16.0	86.7 ± 12.9
	95% CI	(−507.5–245.1)	(−60.2–93.6)	(−187.8–80.2)	(−366.9–274.2)
	Motor Cortex EEG	−311.8 ± 634.7	−95.6 ± 184.4	−99.9 ± 197.1	33.1 ± 75.2
	95% CI	(−176.6–62.6)	(−216.1–19.1)	(−185.7–46.1)	(22.2–88.3)
	NME	0.03 ± 0.01	0.02 ± 0.01	0.02 ± 0.01	0.03 ± 0.01
	95% CI	(0.01–0.04)	(0.01–0.04)	(0.01–0.03)	(0.01–0.03)
	EEG–EMG Coherence	0.06 ± 0.05	0.04 ± 0.02	0.04 ± 0.02	0.06 ± 0.06
	95% CI	(0.02–0.06)	(0.02–0.05)	(0.02–0.03)	(0.02–0.06)
**PLA**	Prefrontal EEG	70.2 ± 22.9	64.9 ± 36.3	80.1 ± 21.1	92.1 ± 11.6
	95% CI	(45.2–96.8)	(2.7–84.3)	(33.1–104.3)	(81.6–102.3)
	Motor Cortex EEG	−27.0 ± 148.9	−80.7 ± 191.9	−10.3 ± 150.3	66.2 ± 57.5
	95% CI	(57.0–88.6)	(−165.7–79.6)	(−109.1–121.1)	(66.4–107.4)
	NME	0.03 ± 0.01	0.02 ± 0.01	0.02 ± 0.01	0.02 ± 0.01
	95% CI	(0.01–0.06)	(0.01–0.05)	(0.01–0.04)	(0.01–0.04)
	EEG–EMG Coherence	0.06 ± 0.06	0.04 ± 0.03	0.09 ± 0.09	0.05 ± 0.05
	95% CI	(0.02–0.06)	(0.02–0.06)	(0.03–0.08)	(0.02–0.06)

NME—neuromuscular efficiency; EEG—electroencephalography; EMG—electromyography; CAF—caffeine; PLA—placebo.

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
