# Peer review of "Caffeine Increased Muscle Endurance Performance Despite Reduced Cortical Activation and Unchanged Neuromuscular Efficiency and Corticomuscular Coherence"

_nutrients, 2019, doi:10.3390/nu11102471_

Round 1

Reviewer 1 Report

The authors present a cross-over study to analyze the effect of 5 mg/kg of caffeine supplement ingestion on muscle endurance performance and changes in cortical and neuromuscular efficiency and EEG-EEM coherence. The authors conducted an interesting paper, however, which needs minor adjustments to be recommended.

Title:

-The acronyms of the title must be removed or explained. 

Abstract:

- The abstract needs to be revised. It is necessary to include more information about the methods (eg caffeine dosage). Also, there is confusion in the use of acronyms. Some of them is not explained before.

Introduction:

- The introduction is too long. In the introduction the authors need to be more objective. It took them to get to the specific topic of the study. The hypothesis proposed are not clearly enough. It would be of interest to better explain the benefits and improvements of measure changes in physical performance with this supplement. It is necessary to justify better the present study. 

- Authors should present in the introduction data from systematic reviews with meta-analyzes

-I think the term muscular endurance may not be the most appropriate to define a task with a duration of 25-30 second. Please, review and amend it.  

Methods:

Please, the warm-up needs to be explained.

Results:

The results do not describe any adverse effect of the caffeine.

95% CI and should be included in the results section. 

Are there any difference between basal and PLA in endurance performance? Can you include the F, p, and ES related with the 7.7% of decrease mentioned in the text? What is the reason of this decrease?

Discussion:

Do you think the daily caffeine ingestion of the sample (3 non-consumers, 5 occasional consumers, 4 daily consumers) can affect the results obtained in your study? Do you analyze the effect of this variable on the results by an statistical analysis? 

Strengths and limitations are not described

The discussion should bring the clinical implication of the finding. What is the practical application of this type of supplementation specifically with this type of task?

Author Response

Dear Dr. Del Coso, J , Editor, Nutrients;

We thank the reviewers for the thoughtful comments and suggestions, they improved the manuscript. We also thank the editor for the opportunity to revise and submit our manuscript (ID nutrients-556319). We made all efforts to incorporate all the suggestions into the manuscript, and a point-by-point letter highlighting our decision is provided below. We hope we have attained the standard required for publication.

Reviewer 1

The authors present a cross-over study to analyze the effect of 5 mg/kg of caffeine supplement ingestion on muscle endurance performance and changes in cortical and neuromuscular efficiency and EEG-EEM coherence. The authors conducted an interesting paper, however, which needs minor adjustments to be recommended.

R: Thank you very much for your comprehensive review.

Title:

-The acronyms of the title must be removed or explained.

R: We removed the acronyms and added EEG-EMG coherence as key-words (as it is a widespread term in exercise sciences scenarios), please let us know if it is ok.

Abstract:

- The abstract needs to be revised. It is necessary to include more information about the methods (eg caffeine dosage). Also, there is confusion in the use of acronyms. Some of them is not explained before.

R: Thank you. We have included more methodological information and removed some acronyms (clarifying others) in 200 words maximum limit.

Introduction:

- The introduction is too long. In the introduction the authors need to be more objective. It took them to get to the specific topic of the study. The hypothesis proposed are not clearly enough. It would be of interest to better explain the benefits and improvements of measure changes in physical performance with this supplement. It is necessary to justify better the present study.

R: We have tried to make it clearer, highlighting the fact that no study aimed to verify if caffeine ingestion affects the coupling between central (CNS) and peripheral (muscles) sites. Regarding the “long introduction”, we attempted to justify every dependent measure from our design. Usually, caffeine studies having mechanistic approaches need a little bigger introduction to properly introduce the research question (justifying all dependent measures) before the hypothesis. We believe this is the case for our study. Thus, we first highlighted the caffeine benefits and likely mechanisms in the first paragraph, then we introduced central (i.e. cortical), peripheral responses (i.e. muscular efficiency) and central-peripheral coupling (EEG-EMG coherence) during the 2nd, 3rd, and 4th paragraphs (broken into 2 paragraphs for a better rational in this new version), respectively, thus saving the last paragraph for the research question and hypothesis. We are concerned in shortening the introduction, as we think that readers will miss important information to understand the study’s rationality. Please, let us know what you think about our amendments. As presented now, the introduction has a length similar to mechanistic studies of caffeine.

- Authors should present in the introduction data from systematic reviews with meta-analyzes

R: inserted, accordingly. We added 2 systematic reviews and meta-analysis about overall caffeine effects on exercise performance in different sport and exercise scenarios. For specific caffeine effects such as in muscular function test and cortical-muscle responses, original studies were used, given that absence of reviews specifically dealing with these effects.

-I think the term muscular endurance may not be the most appropriate to define a task with a duration of 25-30 second. Please, review and amend it. 

R: We have used a definition close to that used by important guidelines (1), defining muscular endurance as light-to-moderate strength exercises (20-60% 1RM) performed for > 15 repetitions (i.e. corresponding to > 20 s of duration). This term has been used by studies approaching similar isometric voluntary contractions (2, 3).

American College of Sports Medicine position stand. Progression models in resistance training for healthy adults. Med Sci Sports Exerc. 2009 Mar;41(3):687-708. doi: 10.1249/MSS.0b013e3181915670. Review.PMID: 19204579 Grgic J1, Mikulic P2. Eur J Sport Sci. 2017 Sep;17(8):1029-1036. doi: 10.1080/17461391.2017.1330362. Epub 2017 May 24. Caffeine ingestion acutely enhances muscular strength and power but not muscular endurance in resistance-trained men. Dunkin JE1, Phillips SM. The Effect of a Carbohydrate Mouth Rinse on Upper-Body Muscular Strength and Endurance. J Strength Cond Res. 2017 Jul;31(7):1948-1953. doi: 10.1519/JSC.0000000000001668.

Methods:

Please, the warm-up needs to be explained.

R: Thank you, added accordingly (unloaded squat, 2 sets of 15 repetitions and 1 min interval between the sets).

Results:

The results do not describe any adverse effect of the caffeine.

R: We added this information, accordingly.

95% CI and should be included in the results section.

R: A table reporting mean (SD) and 95% CI values was included, accordingly (table S1).

Are there any difference between basal and PLA in endurance performance? Can you include the F, p, and ES related with the 7.7% of decrease mentioned in the text? What is the reason of this decrease?

R: Considering your suggestion to compare PLA vs baseline, we reanalyzed the data with ANOVA, having CAF, PLA and baseline as conditions. We had decided to use a paired t-test between CAF and PLA sessions as baseline session was performed in a sequential order i.e., we did not consider baseline when balancing the experiment, as baseline session was used as a familiarization when assessing physiological responses to a “natural” non-supplemented IC exercise. However, given that results did not change in direction, we decided to follow your recommendation. Please see the results section: Condition main effect (F = 8.49, p = 0.002, d = 1.24 very large ES), having CAF a greater time to task failure than baseline (p = 0.006) and PLA (p = 0.007). However, no difference was found between baseline and PLA (p = 1.00) so that the value slightly lower in PLA than baseline may be considered just as variation.

Discussion:

Do you think the daily caffeine ingestion of the sample (3 non-consumers, 5 occasional consumers, 4 daily consumers) can affect the results obtained in your study? Do you analyze the effect of this variable on the results by an statistical analysis?

R: We do not think so. Recent studies support the notion that daily consumption of caffeine has no effect on exercise performance (Gonçalves et al., 2017; Del Coso et al., 2019) so that we have not designed the present study to incorporate this point. As argued in line 99 - 101, we think that responsiveness and habituation were not an issue in our results.

Strengths and limitations are not described

R: Thank you. We complemented the “Methodological aspects” section highlighting some strengths (EEG-EMG coherence) and limitations (caffeine effects on tissue oxygenation and metabolism).

The discussion should bring the clinical implication of the finding. What is the practical application of this type of supplementation specifically with this type of task?

R: please, see amendments included in the “Methodological aspects” section, as mentioned above.

Reviewer 2 Report

Dear Authors,

this study aims to evaluate the mechanism of increased endurance exercise  performance after caffeine ingestion  (by cortical activation or peripheral alterations).

The introduction is clear, with adequate bibliographical references, and summarizes the scientific background and the aims of the authors.

Methods are explained in a complete and structured way. The results are quite well reported, also thanks to the numerous figures.

However, despite the statistical power calculation, it would be appropriate to increase the number of participants to corroborate these results, also to overcome the limit (moreover specified by the authors) of the inter-individual variability of the response to caffeine.

The discussion is wide and well developed.

The study is generally well structured, but of not particular scientific relevance for the obtained results.

Author Response

Dear Dr. Del Coso, J , Editor, Nutrients;

We thank the reviewers for the thoughtful comments and suggestions, they improved the manuscript. We also thank the editor for the opportunity to revise and submit our manuscript (ID nutrients-556319). We made all efforts to incorporate all the suggestions into the manuscript, and a point-by-point letter highlighting our decision is provided below. We hope we have attained the standard required for publication.

Dear Authors,

this study aims to evaluate the mechanism of increased endurance exercise performance after caffeine ingestion  (by cortical activation or peripheral alterations). The introduction is clear, with adequate bibliographical references, and summarizes the scientific background and the aims of the authors. Methods are explained in a complete and structured way. The results are quite well reported, also thanks to the numerous figures. However, despite the statistical power calculation, it would be appropriate to increase the number of participants to corroborate these results, also to overcome the limit (moreover specified by the authors) of the inter-individual variability of the response to caffeine. The discussion is wide and well developed. The study is generally well structured, but of not particular scientific relevance for the obtained results.

R: Thank you very much for your insight. However, we disagree with you about the study’s relevance, as there is only a very few studies simultaneously investigating central and peripheral effects of caffeine on exercise performance (in a controlled setup) [8, 12]. Importantly, this is the first study showing that caffeine effects on CNS were uncoupled with muscle responses, as we found no effects of caffeine ingestion on corticomuscular coupling. The present study contributed to show that caffeine potentiates muscular endurance performance through central rather than peripheral effects and coupled CNS-muscle responses. EEG-EMG coherence is insightful to understand how cortical responses play a role on the control of movement. Thus, this may be of value for exercise performance and clinical scenarios, as one may want to focus on CNS alterations without altering CNS-muscle coupling or muscle responses. Regarding the number of participants, we have estimated the minimal sample size necessary to reveal any (if present) effect of caffeine ingestion. Once we got it, we decided to finalize the data sampling as the progression of a research that has already reached its expected results would implicate in ethical and financial aspects.

Bowtell, J.L.; Mohr, M.; Fulford, J.; Jackman, S.R.; Ermidis, G.; Krustrup, P.; Mileva, K.N. Improved Exercise Tolerance with Caffeine Is Associated with Modulation of both Peripheral and Central Neural Processes in Human Participants. Front. Nutr. 2018, 5, 6. Plaskett, C.J.; Cafarelli, E. Caffeine increases endurance and attenuates force sensation during submaximal isometric contractions. J Appl Physiol 2001, 91, 1535–1544